# A Novel Muscle Atrophy Mechanism: Myocyte Degeneration Due to Intracellular Iron Deprivation

**DOI:** 10.3390/cells11182853

**Published:** 2022-09-13

**Authors:** Dae Keun Suh, Won-Young Lee, Woo Jin Yeo, Bong Soo Kyung, Koo Whang Jung, Hye Kyung Seo, Yong-Soo Lee, Dong Won Suh

**Affiliations:** 1Department of Orthopedic Surgery, Korea University Anam Hospital, 73 Goryeodae-ro, Seongbuk-gu, Seoul 02841, Korea; 2Hand and Foot Center, Barunsesang Hospital, #75-5, Yatap-ro, Seongnam-si 13497, Gyeonggi-do, Korea; 3Joint Center, Barunsesang Hospital, #75-5, Yatap-ro, Seongnam-si 13497, Gyeonggi-do, Korea; 4Research Center for Cartilage Regeneration, Barunsesang Hospital, #75-5, Yatap-ro, Seongnam-si 13497, Gyeonggi-do, Korea

**Keywords:** muscle atrophy, iron, hypoxia-inducible factor 1, ferroportin, myostatin, transcription factor

## Abstract

Muscle atrophy is defined as the progressive degeneration or shrinkage of myocytes and is triggered by factors such as aging, cancer, injury, inflammation, and immobilization. Considering the total amount of body iron stores and its crucial role in skeletal muscle, myocytes may have their own iron regulation mechanism. Although the detrimental effects of iron overload or iron deficiency on muscle function have been studied, the molecular mechanism of iron-dependent muscle atrophy has not been elucidated. Using human muscle tissues and in the mouse rotator cuff tear model, we confirmed an association between injury-induced iron depletion in myocytes and muscle atrophy. In differentiated C2C12 myotubes, the effects of iron deficiency on myocytes and the molecular mechanism of muscle atrophy by iron deficiency were evaluated. Our study revealed that the lower iron concentration in injured muscle was associated with the upregulation of ferroportin, an iron exporter that transports iron out of cells. Ferroportin expression was increased by hypoxia-inducible factor 1α (HIF1α), which is activated by muscle injury, and its expression is controlled by HIF1 inhibitor treatment. Iron deprivation caused myocyte loss and a marked depletion of mitochondrial membrane potential leading to muscle atrophy, together with increased levels of myostatin, the upstream regulator of atrogin1 and muscle RING-finger protein-1 (MuRF1). Myostatin expression under iron deficiency was mediated by an orphan nuclear receptor, dosage-sensitive sex reversal-adrenal hypoplasia congenita critical region on the X chromosome (DAX1).

## 1. Introduction

Skeletal muscle atrophy is characterized by progressive degeneration or shrinkage of myocytes due to an imbalance between anabolic and catabolic processes [1]. Various atrophy-associated factors and their molecular mechanisms have been identified in the context of muscle degeneration, including aging, injury, inflammation, cancer, cachexia, mechanical unloading, and metabolic disorders [2,3,4,5]. With respect to skeletal muscle atrophy, the ubiquitin–proteasome system (UPS) is representative of the three major protein degradation pathways, the others being the autophagy–lysosome pathway and apoptosis [6,7]. In the UPS, targeted proteins are ubiquitinated by ubiquitin–protein ligases, and the tagged proteins are then recognized by the 26S proteasome, which is responsible for protein degradation [8]. UPS is known to be closely associated with the muscle-specific E3 ligases muscle atrophy F-box (MAFbx)/atrogin1 and muscle RING finger 1 (MuRF1) [9,10]. These muscle-specific E3 ubiquitin ligases are increased transcriptionally in skeletal muscle under atrophy-inducing conditions such as those mentioned above, and certain myofibrillar proteins and myogenic proteins have been identified as their target substrates [9].

Iron is essential for oxygen (O_2_) transport, DNA synthesis, metabolic energy, and cellular respiration in most organisms [11]. Most of the iron in vertebrates is found as hemoglobin in red blood cells, with the remainder generally stored in muscle, liver, and macrophages [12]. In plasma, transferrin-bound iron is transported to iron-requiring tissues. Iron is mostly sourced from that recycled during senescent red blood cell removal by macrophages; the rest is absorbed from food in the duodenum and transferred from cells to plasma by the transmembrane iron transporter ferroportin [13]. As disrupted iron homeostasis is closely associated with many diseases, its fine-tuned regulation is very important. The liver functions as the pivotal regulator of iron homeostasis, and the main synthesizer of hepcidin. Once hepcidin binds to ferroportin in enterocytes, macrophages, and hepatocytes, the complex is internalized and degraded, thereby inhibiting iron export from cellular stores. Hepcidin expression is therefore regulated in line with plasma iron levels to avoid excess or deficiency [14].

As an iron-containing O_2_-storage protein, myoglobin plays a pivotal role as an O_2_ provider to myocytes during hypoxic conditions or to mitochondria during increased metabolic activity [15], and is enriched in muscle. Studies of the relationship between muscle iron metabolism and muscle physiology have reported that iron deficiency impairs skeletal muscle functioning in the context of mitochondrial functions during energy metabolism [16,17,18]. Conversely, excess iron accumulation in muscles triggers iron-dependent oxidative stress, producing reactive oxygen species (ROS) [19,20,21], indicating another route by which iron imbalance can lead to skeletal muscle atrophy. However, the full process and exact molecular mechanisms of iron imbalance-induced muscle atrophy remain to be elucidated. In addition, local iron metabolism in myocytes and its crosstalk with muscle physiology remain poorly understood.

In this study, we examined any local mechanisms orchestrating iron metabolism within myocytes in the context of muscle injury in humans as well as in animal models, and searched for the potent molecule(s) involved in iron regulation and muscle physiology.

## 2. Materials and Methods

### 2.1. Subjects and Human Tissue Acquisition

From the available patients due for surgical treatment for open reduction and internal fixation (ORIF) in the hand and rotator cuff repair (RCR) in the shoulder, we selected 30 without any prior surgical history at that location or steroid injection during the 3 months before surgery. In the ORIF group (n = 15), small parts of the torn pronator quadratus muscle (5 × 5 mm) were collected from the torn area (injured) by a fractured bone fragment, and from nearby but intact muscle (control) (Figure 1a,b). In the RCR group (n = 15), limited to patients with medium-sized rotator cuff tears (1–3 cm tear size), torn supraspinatus muscle (injured) and intact deltoid muscle (control) samples were acquired: supraspinatus muscle [5 × 5 mm] 1 cm from the musculotendinous junctions; and deltoid muscle [5 × 5 mm] from the anterior portion of the deltoid at footprint level, using an arthroscopic punch through the lateral portal during arthroscopic rotator cuff repair (Figure 1c,d). Ethical issues required use of the intact deltoid, not supraspinatus, muscle as the control. The collected muscle samples were immediately frozen at −80 °C for RT-qPCR and Western blot analyses or were fixed in fresh 4% buffered formalin for 24 h at 4 °C for tissue histology.

### 2.2. Mouse Muscle Injury Model

We generated a mouse rotator cuff tear model, as described previously [22]. Briefly, male, 8-week-old C57BL/6 mice were randomly divided into four groups (to be sacrificed 1, 2, 4, and 6 weeks after RCT; n = 15 per group) and subjected to unilateral complete supraspinatus tendon transection under anesthesia. After sacrifice, the supraspinatus muscles were completely separated from the scapular fossa. The muscle samples were used for total RNA or protein extraction (n = 12 per group) and histological analyses (n = 3 per group). The contralateral supraspinatus muscle was used as a control.

### 2.3. Measurement of Iron Levels 

Iron was measured using an iron colorimetric assay kit (K390-100; BioVision, Milpitas, CA, USA), in accordance with the manufacturer’s instructions. Muscle tissue or intracellular iron concentration was measured from tissue homogenate or harvested cells, respectively.

### 2.4. Reverse Transcription Quantitative Real-Time PCR (RT-qPCR) Analysis

Total RNA was extracted from the dissected muscle tissues or cultured cells using TRIzol reagent (Molecular Research Center, Cincinnati, OH, USA) and used for cDNA synthesis. A CFX Connect Real-Time System (Bio-Rad, Hercules, CA, USA) was used for qPCR, with qPCRBIO SyGreen Blue Mix Lo-ROX (PCR Biosystems, London, UK). All data were normalized to β-actin expression and the data were quantitatively analyzed. The primer sequences used are shown in Table 1. All experiments were repeated at least three times.

### 2.5. Western Blot Analysis

Homogenized muscle tissues or cultured cells were lysed with radioimmunoprecipitation assay lysis buffer (RC2002-050-00, Biosesang, Gyeonggi-do, Republic of Korea), separated using 10% sodium dodecyl sulfate polyacrylamide gel electrophoresis, and transferred to a polyvinylidene fluoride membrane (IPVH00010, Merck, Darmstadt, Germany). The membranes were probed with each corresponding antibody: anti-myosin heavy chain (MYH) (sc-376157, Santa Cruz Biotechnology, Dallas, TX, USA), anti-fatty acid binding protein 4 (FABP4) (NBP1-89218, Novus Biologicals, Littleton, CO, USA), anti-ferritin heavy chain (sc376594, Santa Cruz Biotechnology), anti-ferroportin (NBP1-21502, Novus Biologicals), anti-hypoxia-inducible factor 1α (HIF1α) (ab82832, Abcam, Cambridge, MA, USA), anti-N-Myc downstream regulated 1 (NDGR1) (sc398291, Santa Cruz Biotechnology), anti-myogenic factor 5 (Myf5) (sc518039, Santa Cruz Biotechnology), anti-myoblast determination protein 1 (MyoD1) (ab133627, Abcam), anti-myogenin (ab1835, Abcam), anti-Fbx32/atrogin1 (ab168372, Abcam), anti-MuRF1 (AF5366, R&D Systems, Minneapolis, MN, USA), anti-myostatin (ab203076, Abcam), anti-α-tubulin (ab7291, Abcam), and anti-glyceraldehyde 3-phosphate dehydrogenase (sc32233, Santa Cruz Biotechnology). Immunoreactive proteins were assessed using an LAS-3000 image analyzer (Fuji Film, Tokyo, Japan). The proteins were quantified by densitometry using ImageJ software (Version 1.8.0, National Institutes of Health, Bethesda, MD, USA). 

### 2.6. Tissue Histology

Paraffin-embedded tissue sections (5 μm) were cut in the cross-sectional plane at the mid-portion of the muscle, deparaffinized in xylene, and rehydrated through an ethanol/water series. Tissue slides were then subjected to H&E staining. All slides were evaluated under an Eclipse Ni-U microscope (Nikon, Tokyo, Japan), and images were acquired with a DS-Ri1 camera (Nikon) and analyzed using NIS Elements F4.00.00 version 4.0 (Nikon). For immunofluorescence microscopy, fixed frozen muscle sections (0.5 μm) were labeled with an anti-ferroportin antibody, and the resulting immune complexes were visualized with an Alexa Fluor 488 goat anti-rabbit IgG H&L (ab150077, Abcam) secondary antibody. Nuclei were stained with 4′,6-diamidino-2-phenylindole (Vectashield Hardset Antifade Mounting Medium, H-1500, Vector Laboratories, Burlingame, CA, USA). Images were acquired using an upright fluorescence microscope (BX61-32FDIC, Olympus, Tokyo, Japan).

### 2.7. Cell Culture, Reagents, Hypoxic Incubation, Transient Transfection and Adenoviral Infection

C2C12 mouse myoblasts were cultured in Dulbecco’s modified Eagle’s medium (DMEM) supplemented with 10% fetal bovine serum and antibiotics at 37 °C in a humidified atmosphere containing 5% CO_2_. For differentiation into myotubes, C2C12 cells were maintained in DMEM supplemented with 2% horse serum for 72 h, and the medium was replaced every other day. The majority of C2C12 cells were differentiated into myotubes after 72 h in differentiation medium, although there were some residual myoblasts in the area of low cell density. For the hypoxic challenge, cells were incubated in a hypoxic chamber (0.1% O_2_, 5% CO_2_, 10% H_2_, 85% N_2_) for the designated period. The reagents used for cell treatment were lipopolysaccharide (LPS) (*E. coli* 026:B6, L2654, Merck), interleukin (IL)-1β (Cell Signaling Technology, Danvers, MA, USA), IL-6 (CYT-213, PROSPEC, Hamada St. 8, Rehovot 7670308, Israel), tumor necrosis factor-α (TNF-α) (Cell Signaling Technology), HIF1 inhibitor (InSolution^TM^ HIF-1 inhibitor, Cat# 400092; Calbiochem, San Diego, CA, USA), deferoxamine (DFO) (D9533, deferoxamine mesylate salt, Merck), and FeSO_4_ (F7002, Iron(II) sulfate heptahydrate, Merck). Transient transfections were performed as previously described using Lipofectamine 2000 (Thermo Fisher, Waltham, MA, USA). The DNA constructs used for overexpression of the nuclear receptor were sourced from Chung-Ang University (Seoul, Republic of Korea), pcDNA3-HIF-1α from University of Utah School of Medicine (Salt Lake City, UT, USA), and the adenovirus expressing DAX1 from the School of Biological Sciences and Technology (Chonnam National University, Gwangju, Republic of Korea).

### 2.8. Cell Viability Assay and Flow Cytometric Analysis

The cell viability of the differentiated C2C12 cells (myotubes) was assessed using the 3-(4,5-dimethylthiazol-2-yl)-2,5-diphenyltetrazolium bromide (MTT, M2128, Merck) assay. C2C12 myotubes were plated at a density of 1 × 10^4^ cells/well in 24-well culture plates and incubated for 24 h at 37 °C and 5% CO_2_. Subsequently, the culture medium was replaced with fresh DMEM or DMEM containing different concentrations of DFO or FeSO_4_. At 48 h after reagent treatment, MTT (5 mg/mL) was added, and the culture plates were incubated at 37 °C for 4 h. The resulting formazan product was dissolved in dimethyl sulfoxide, and its absorbance measured at 560 nm using an Ultra Multifunctional Microplate Reader (Tecan US, Morrisville, NC, USA). Cell viability was determined from the readings using the formula:% Viability = (fluorescence value of MSM/fluorescence value of untreated control) × 100

All experiments and measurements were performed in triplicate. For flow cytometric analysis, the cells were plated at a density of 1 × 10^5^ cells/well in 6-well plates and incubated for 24 h. The cells were then treated with different doses of DFO or FeSO_4_ in DMEM for an additional 48 h. The harvested cells were washed twice with phosphate-buffered saline (PBS) and resuspended in PBS (0.5 mL). The size and granularity of cells were determined by forward and side scatter (FSC and SSC) gating, respectively, during flow cytometry on a FACScalibur instrument, and analyzed with CellQuest software (Version 3.3, BD Bioscience, San Jose, CA, USA). The uptake of 3,3′-dihexyloxacarbocyanine iodide was analyzed to measure the loss of mitochondrial membrane potential, reflecting initiation of the pro-apoptotic signal. Flow cytometric DNA histograms were analyzed using the FL2 dot plot, representing the proportions of cells in the sub G1, G0/G1, S, and G2/M phases of the cell cycle.

### 2.9. Statistical Analyses 

Data are expressed as mean ± standard error. Differences between the control and experimental groups were assessed using *t*-tests (α = 0.05) in GraphPad Prism 5.01 (GraphPad Software, La Jolla, CA, USA). Differences were considered significant when *p* values were <0.05.

## 3. Results 

### 3.1. Decreasing Iron Levels Correlate with Increasing Ferroportin Expression in the Injured Muscles

We examined the correlation between muscle atrophy and myocyte iron concentration by comparing iron levels in torn (injured) muscle and intact (control) muscle. Histological analysis revealed remarkable inflammation, cell shrinkage, and a reduced myocyte population in the injured muscle, indicating progressive muscle degeneration (Figure 2a). The myocyte iron concentration was significantly lower in the injured muscle compared to that in the intact muscle (Figure 2b). In this context, markedly higher ferroportin mRNA expression in the injured muscle was noteworthy. Injured muscle also expressed the primary intracellular iron-storage protein, ferritin, at lower levels, while its hepcidin mRNA levels were unchanged from their low basal expression levels, suggesting that hepcidin is not a major iron regulator in skeletal muscle (Figure 2c). Western blot analysis also showed significantly higher ferroportin protein expression in the injured muscle, and lower expression of myosin heavy chain and higher fatty acid-binding protein 4, the key muscle degeneration markers (Figure 2d). Immunofluorescence analysis showed that ferroportin protein expression was markedly higher in the cell membrane of the torn supraspinatus muscle compared to that in the intact deltoid muscle (Figure 2e). These results indicate that decreased iron levels due to muscle injury are associated with the upregulation of ferroportin expression in myocytes.

### 3.2. Ferroportin Expression Is Regulated by Hypoxia, Leading to Lower Iron Concentration

Regarding the molecular mechanism underlying ferroportin induction by muscle injury, we first examined the effect of inflammation-associated molecules as an acute inflammatory response occurred immediately following injury (Figure 3a). Excluding the positive control, the iron supplement FeSO_4_, levels of the pro-inflammatory molecules LPS, IL-1β, IL-6, and TNF-α did not affect ferroportin gene expression, suggesting that these inflammatory cytokines are not responsible for ferroportin induction. Thus, we speculated that ferroportin expression could be regulated by hypoxia, because muscle injury leads to hypoxic conditions [23]. We therefore examined ferroportin expression in differentiated C2C12 myotubes cultured under low-oxygen conditions. Ferroportin mRNA expression was significantly increased by hypoxia in a time-dependent manner (Figure 3b). In addition, overexpression of HIF1α significantly induced ferroportin gene expression, while its expression was suppressed by HIF1 inhibitor treatment (Figure 3c). Accordingly, Western blot analysis also revealed the regulation of ferroportin expression by HIF1α at the protein level (Figure 3d). To examine the correlation between ferroportin and HIF1α expression in vivo, we used the mouse model described above. Histological images showed an inflammatory response and ectopic fatty infiltration in the injured muscle, which are representative of muscle atrophy (Figure 3e). Expression levels of both HIF1α and ferroportin mRNA and protein were significantly higher in the injured muscle (Figure 3f,g). Interestingly, myocyte iron levels in injured muscle were markedly lower, suggesting that injury-induced iron reduction in muscle is caused by ferroportin upregulation (Figure 3h). The induction of ferroportin mRNA was sustained during the acute phase of muscle injury (Figure 3i). Taken together, these results suggest that muscle injury-induced hypoxic conditions regulate ferroportin expression through direct activation of HIF1α, and increased ferroportin expression leads to iron reduction in myocytes by exportation of intracellular iron.

### 3.3. Iron Deprivation Leads to Smaller Cell Population and Myocyte Mitochondrial Dysfunction

In addition to being an essential micronutrient for various cellular events, oxygen-supplying iron in myoglobin plays an important role in skeletal muscle functions, such as oxidative energy metabolism. Above all, skeletal muscle mitochondrial respiration relies heavily on iron availability. To determine whether defective iron homeostasis in myocytes correlates directly with muscle physiology, we challenged C2C12 myotubes with an iron chelator, DFO, or an iron supplement, FeSO_4_. DFO challenge produced significant cytotoxicity even at a relatively low concentration (100 μM), while FeSO_4_ treatment produced noticeable cell damage only at the highest dosage (400 μM) (Figure 4a). Interestingly, the decreased cell viability by iron deficiency was significantly recovered by iron supply with concomitant FeSO_4_ treatment, suggesting that iron deficiency causes less cell viability (Figure 4b). Chelation by DFO was clearly evidenced by a markedly lower cellular iron concentration (Figure 4c). Corroborating these results, DFO-treated differentiated myotubes were significantly smaller, and the cell population smaller compared with the controls (Figure 4d). Flow cytometry analysis also revealed a remarkable decrease in the FSC due to cell shrinkage and an increase in SSC, probably due to cell death, leading to the smaller cell population under iron deprivation (Figure 4e). In addition, these conditions also significantly depleted mitochondrial membrane potential, reflecting the myocytes’ intracellular iron deficiency (Figure 4f). Iron deprivation also led to significant cell cycle arrest, revealing a higher proportion of cells in the G0/G1 (M2) phase and a lower one in the S (M3) and G2/M (M4) phases, indicating cell death (Figure 4g). Interestingly, these cellular events were unaffected by FeSO_4_ supplementation, suggesting that iron deficiency is more harmful to myocytes than iron overload. Taken together, these results suggest that iron deficiency is a substantial factor in mitochondrial membrane potential and myocyte population reduction.

### 3.4. Iron Deficiency in Myocytes Is Accompanied by Increased Expression of Atrophy-Associated Genes

To examine whether iron deficiency in myocytes is directly related to muscle atrophy, we measured the expression of muscle-associated molecules in iron-deprived and iron-challenged myocytes. Iron deprivation caused the expected significantly lower gene expression levels for ferroportin and ferritin, while an iron-regulated gene, N-Myc downregulated gene 1 (NDRG1) is markedly upregulated by cellular iron depletion (Figure 5a). Among the muscle atrophy-associated genes, mRNA levels for the representative muscle-specific E3 ligases, atrogin1 and MuRF1, were significantly higher under iron deprivation, while those for the myogenic factors Myf5, MyoD, and myogenin were lower. Interestingly, the expression of myostatin, a growth differentiation factor of the transforming growth factor β (TGF-β) family, was dramatically higher in iron-deprived myocytes (Figure 5b). Western blot analysis corroborated the qPCR data (Figure 5c–e). Although atrogin1, MuRF1, and myostatin comprise the critical factors for muscle atrophy, we focused specifically on myostatin because it is not only the molecule with the highest iron deficiency-induced expression level, but also an upstream regulator of atrogin-1 and MuRF1 [24].

Myostatin mRNA expression was significantly higher under iron deprivation, while iron supplementation tended to lower it (Figure 5f). In particular, the result showing that iron deficiency-induced myostatin expression was dramatically reduced by iron supply suggests an obvious iron-dependency of myostatin expression (Figure 5g). Taken together, these results indicate that myostatin, as an upstream regulator of atrogin1 and MuRF1, is a substantial target molecule for muscle atrophy caused by iron deprivation in myocytes.

### 3.5. Iron Deficiency-Induced Myostatin Expression Is Mediated by an Orphan Nuclear Receptor, DAX1

To identify the transcription factor directly involved in myostatin expression within myocytes, we performed a transient transfection assay with various nuclear receptors as transcription factors in C2C12 myotubes. Among the tested nuclear receptors, an orphan nuclear receptor, dosage-sensitive sex reversal-adrenal hypoplasia congenita critical region on the X chromosome (DAX1) was the most potent such transcription factor among those tested (Figure 6a). DAX1 gene expression was significantly upregulated by iron deprivation, but lower under iron supplementation (Figure 6b). Like myostatin expression, iron deficiency-induced DAX1 expression was dramatically lower under iron supplementation, revealing its iron dependency (Figure 6c). Adenoviral overexpression of DAX1 induced significantly higher mRNA expression of myostatin, atrogin1, and MuRF1 (Figure 6d). Western blot analysis revealed that myostatin, atrogin1, and MuRF1 protein levels were remarkably augmented by DAX1 overexpression (Figure 6e).

## 4. Discussion

Given the important roles of iron in oxygen storage in myoglobin and in oxidative phosphorylation-mediated ATP synthesis by mitochondria, muscle-specific regulation of iron homeostasis may occur, such that its disruption can induce impaired muscle function. In this study, using muscle tissue from human and animal models, we confirmed that damage-induced hypoxic conditions increase the expression of ferroportin, a representative iron metabolism regulator, in myocytes. Upregulated ferroportin expression caused intracellular iron deficiency, leading to mitochondrial dysfunction and muscle atrophy mediated by the induction of myostatin, a key regulator of atrogin1 and MuRF1 expression. Although no molecular mechanism was evident, ferroportin-induced iron deficiency increased the expression of DAX1, an orphan nuclear receptor in myocytes, and DAX1 was found to be a novel transcriptional regulator of myostatin expression.

Systemic iron homoeostasis has been extensively studied in the liver, but the muscle-specific local regulation of iron homeostasis has not been fully elucidated. Our study demonstrated that muscle injury-induced hypoxic conditions lead to significant ferroportin expression, causing iron deficiency in myocytes. It has been proposed that hypoxic conditions induce ferroportin expression in various tissues [25,26], supporting our results. In general, systemic iron homeostasis is controlled by a key iron regulatory hormone, hepatic hepcidin [27]. However, hepcidin expression did not differ in the intact and injured muscles (Figure 2c). Moreover, basal hepcidin expression levels were relatively low compared to those of ferroportin (threshold cycle values 29–32 and 23–26, respectively), suggesting that ferroportin predominates in skeletal muscle. This is consistent with previous studies showing that hepcidin and its regulator hemojuvelin are not critical for muscle iron homeostasis, rather they are primarily important in the liver [28,29]. Regarding ferroportin expression, we could not rule out the possibility of different cell origins, such as muscle-infiltrating macrophages, for its expression [30]. However, the obvious protein expression of ferroportin in C2C12 myotubes following HIF1α treatment, and results from other studies, demonstrate ferroportin expression in myocytes [21,28]. While the major mechanism in systemic iron homeostasis is hepcidin-dependent regulation, local regulation of iron appears to be mediated by extracellular stimuli such as hypoxia, and rapidly and beneficially determines the fate of challenged cells or tissues.

Skeletal muscle is a metabolically active tissue due to its high energy requirements, and oxidative metabolism occurs mainly in myocyte mitochondria. Mitochondria are enriched in skeletal muscle and play a key role in a variety of cellular functions, including oxidative phosphorylation-mediated ATP synthesis. Iron is essential to the electron transport chain, enabling cellular respiration and subsequent ATP production. Iron imbalance has a serious, detrimental effect on mitochondrial function. In the present study, iron deficiency following chelation with DFO led to significantly defective mitochondrial membrane potentials (Figure 4e), supporting a previous study demonstrating that iron deficiency-induced loss of skeletal muscle is caused by increased mitochondrial clearance [18]. Although the cause and effect chain was unclear, intracellular iron deficiency in myocytes led to remarkable myocyte shrinkage and cell cycle arrest, indicating an apoptotic tendency (Figure 4d,g). It is well known that loss of mitochondrial transmembrane potential is closely associated with apoptosis [31]. Thus, intracellular iron deficiency induces decreased skeletal muscle mitochondrial quantity and function, resulting in impaired oxidative substrate metabolism and muscle loss. 

Myostatin, also known as growth/differentiation factor-8, is a member of the TGF-β family, and plays a crucial role in the negative regulation of muscle mass. Beginning with the binding of activin receptors, myostatin-mediated signaling activates FoxO, leading to the upregulation of the proteasome ubiquitin ligases MuRF1 and atrogin1, which participate in protein degradation [32]. We demonstrated that intracellular iron deficiency in myocytes is associated with the highest expression levels of myostatin, atrogin-1, and MuRF1, which comprise the critical factors for muscle atrophy. A variety of regulators of myostatin expression have been reported at the epigenetic, transcriptional, post-transcriptional, and post-translational levels [33]. Interestingly, we identified DAX1 as a novel transcription factor for myostatin expression. DAX1 was initially identified as part of a contiguous gene syndrome and is known to function in the proper formation of the adult adrenal gland to maintain the steroidogenic axis. However, additional functional roles, such as in embryonic development and the maintenance of embryonic stem cell pluripotency, have recently been reported [34]. Although DAX1 is well known as a transcriptional repressor, it can also regulate its target gene by binding with other co-repressors, such as the nuclear receptor co-repressor and silencing mediator of retinoid–thyroid receptor [34]. Thus, the transcriptional activation of myostatin by DAX1 could be mediated by binding with an unknown co-repressor on the promoter of myostatin. With respect to its expression in skeletal muscle, it is possible that DAX1 is upregulated in differentiated stem cells within muscle tissue by stimuli such as iron deficiency, given that DAX1 expression is closely associated with stem cell differentiation or tumorigenesis [35,36,37]. To better understand the essential role of DAX1 in myocytes, a myocyte-specific gain or loss of function strategy may be needed.

In conclusion, our study revealed that muscle injury induces intracellular iron deprivation through induction of iron export in a hypoxic environment, and that iron deficiency in myocytes causes muscle loss or atrophy by upregulation of myostatin, leading to atrogin1 and MuRF1 expression. We also demonstrated for the first time that the nuclear receptor DAX1 is a regulator of myostatin expression in iron-deficient myocytes (Figure 7). Our present study strongly supports the conclusion that iron deficiency is associated with skeletal muscle dysfunction, as reported previously [38,39]. Our novel findings will also support a better understanding of the pathways contributing to injury-induced muscle atrophy or loss of muscle mass via regulation of intracellular iron. In this context, identifying the specific inhibitor(s) of DAX1, a potent regulator of myostatin expression, could be useful in clinical applications. 

## Figures and Tables

**Figure 1 cells-11-02853-f001:**
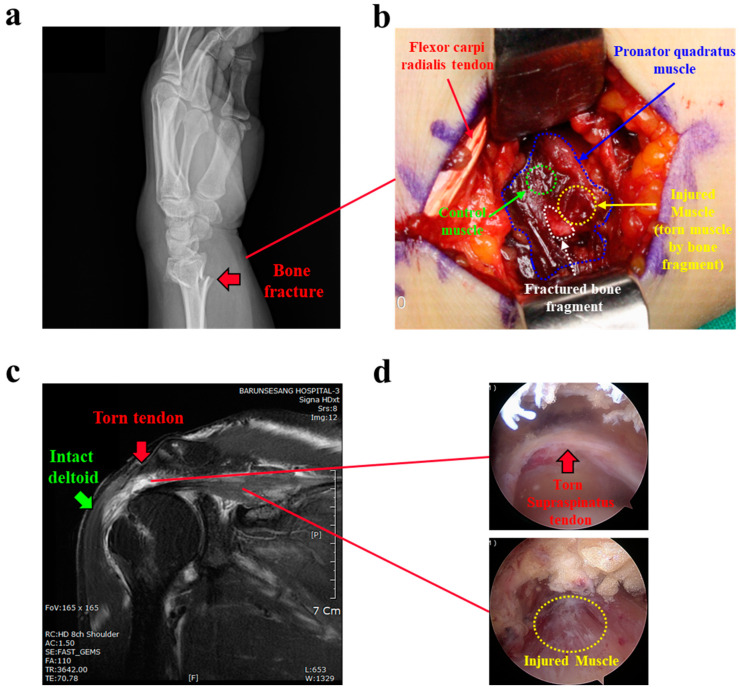
Human muscle tissue preparation (**a**) A representative X-ray image of a forearm wrist bone fracture. Red arrow indicates distal radius fracture. (**b**) Open reduction and internal fixation surgery for the patient in panel (**a**). Torn pronator quadratus muscle (yellow dotted area) beside a fractured bone fragment and intact muscle (green dotted area) separate from the injured area were sampled as injured and control muscles, respectively (n = 15 per group). (**c**) A representative MRI image of rotator cuff tear. Red arrow indicates torn supraspinatus tendon (white area), and green arrow indicates intact deltoid muscle. (**d**) Arthroscopic rotator cuff repair surgery for the patient in panel (**c**). Upper panel shows the torn supraspinatus tendon of rotator cuff in shoulder. Lower panel shows the injured supraspinatus muscle (yellow dotted area). Intact deltoid muscles were acquired arthroscopically during surgery and used as a control muscle (n = 15 per group).

**Figure 2 cells-11-02853-f002:**
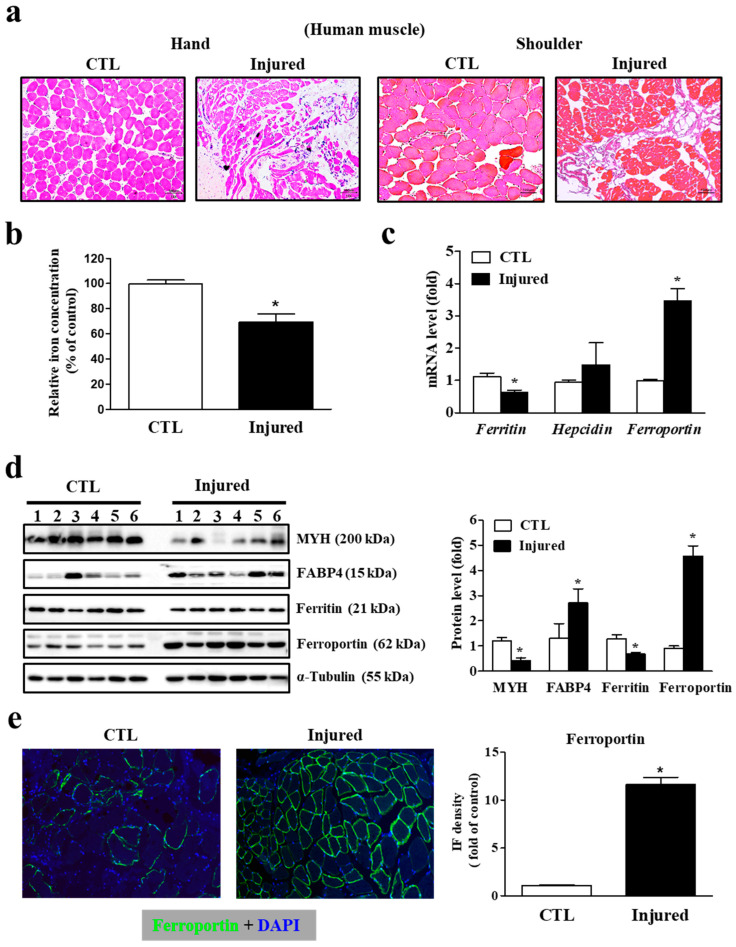
Correlation between reduced iron levels and increased ferroportin expression in injured muscles. (**a**) Histological difference between intact control (CTL) muscle and injured muscle by hematoxylin and eosin staining. Pictures are representative of the muscle tissue from pronator quadratus (hand) and rotator cuff (shoulder) muscles (n = 10 per group). Scale bar: 100 μm. (**b**) Intracellular iron concentration between intact control and injured muscles (n = 30 per group). (**c**) mRNA expression of ferritin, hepcidin and ferroportin for intact control and injured muscles (n = 30 per group). (**d**) Western blot analyses for myosin heavy chain (MYH), fatty acid binding protein 4 (FABP4), ferritin, ferroportin and α-tubulin protein expression in intact control and injured muscles (n = 6 per group). A densitometric analysis of the Western blots is shown in the graph on the right. (**e**) Immunofluorescence (IF) microscopy images showing ferroportin expression. The blue signals indicate cell nuclei. Images are representative of independent tissue samples (n = 10 per group). Scale bar: 100 μm. The negative control (secondary antibody only) for the IF showed no specific signal (data not shown). Ferroportin expression levels (green staining) were assessed with densitometry using Image J software (Version 1.8.0, National Institutes of Health, Bethesda, MD, USA), shown on the right. Data represent mean ± SEM. * *p* < 0.05.

**Figure 3 cells-11-02853-f003:**
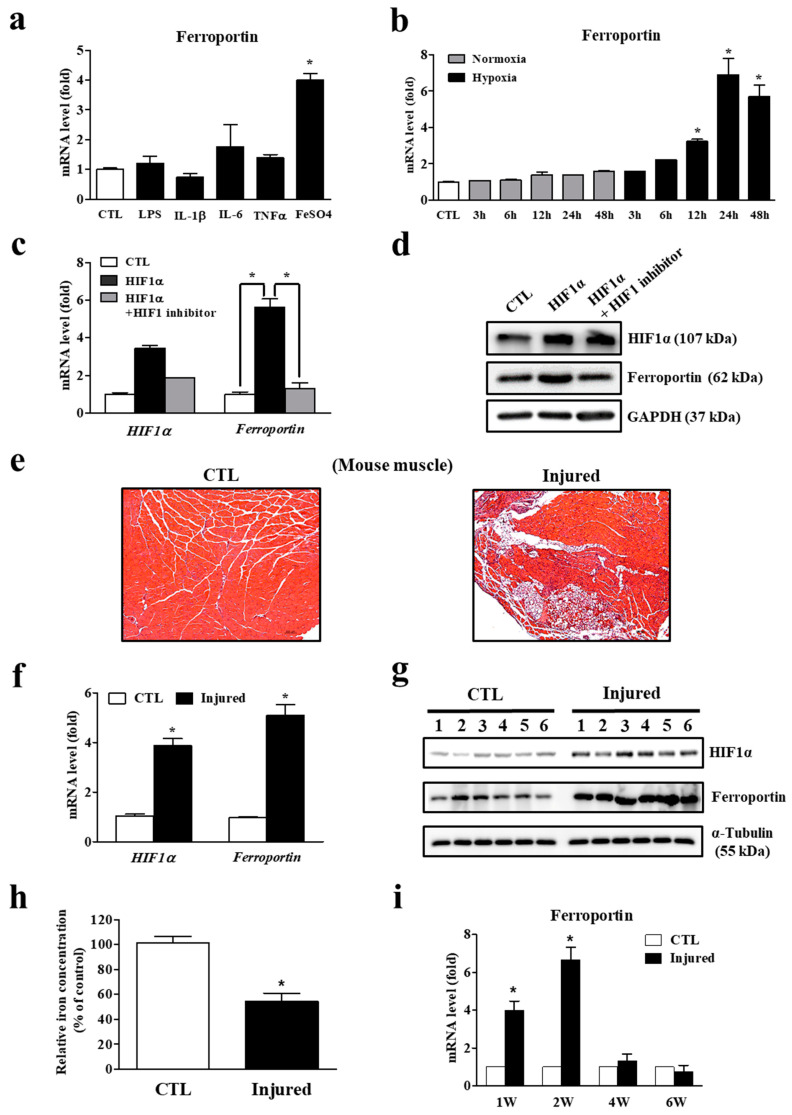
Ferroportin expression by hypoxia and decrease in iron levels. (**a**) Expression of ferroportin mRNA in C2C12 myotubes treated with lipopolysaccharides (LPS, 1 μg/mL), interleukin 1β (IL-1β, 100 ng/mL), IL-6 (100 ng/mL) and tumor necrosis factor α (TNF-α, 100 ng/mL) for 24 h. FeSO_4_ (400 μM/mL) was used as a positive control for ferroportin induction. (**b**) The expression of ferroportin mRNA in C2C12 myotubes after hypoxic challenge for the periods shown. (**c**) Expression of hypoxia-inducible factor 1 alpha (HIF1α) and ferroportin mRNA in C2C12 myotubes by overexpression of HIF1α in the presence or absence of HIF1 inhibitor (20 μM) for 48 h. (**d**) Expression of HIF1α and ferroportin protein in C2C12 myotubes by overexpression of HIF1α in the presence or absence of HIF1 inhibitor for 48 h. (**e**) Histological comparison of the intact supraspinatus and injured supraspinatus muscle in mice after rotator cuff tear by hematoxylin and eosin staining. Images are representative of independent tissue samples (n = 3 per group). Scale bar: 50 μm. (**f**) Expression of HIF1α and ferroportin mRNA in control and injured muscles after rotator cuff tear (RCT) (n = 12 per group). (**g**) Western blot analyses for HIF1α and ferroportin protein expression between control and the injured muscle after RCT (n = 6 per group). (**h**) Iron levels in control and injured muscles after RCT (n = 12 per group). (**i**) Ferroportin mRNA expression changes in supraspinatus muscle after rotator cuff tear in mice (n = 12 per group). Data represent mean ± SEM. * *p* < 0.05. CTL: control.

**Figure 4 cells-11-02853-f004:**
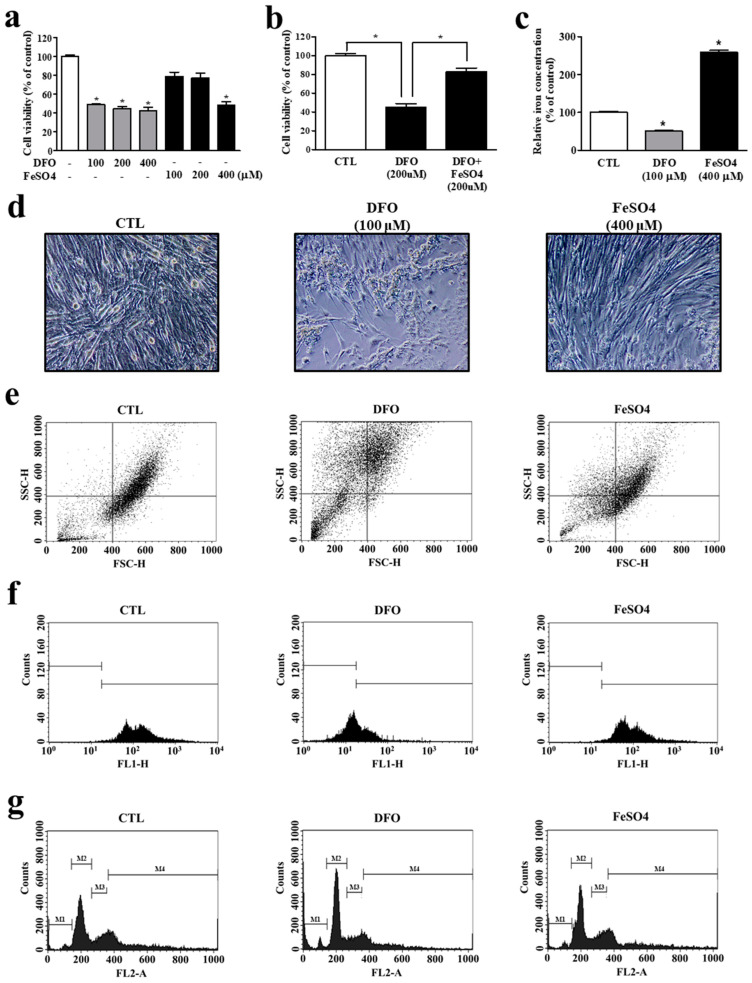
Reduced cell population and mitochondrial dysfunction following iron deprivation in myocytes. (**a**) Cell viability of C2C12 myotubes treated with varying doses of deferoxamine (DFO) and FeSO_4_ for 48 h. (**b**) Recovered cell viability by iron supply with FeSO_4_. (**c**) Intracellular iron concentration in C2C12 myotubes treated with DFO and FeSO_4_. (**d**) Morphology of C2C12 myotubes following DFO and FeSO_4_ treatment. (**e**) Flow cytometry analysis of C2C12 myotubes treated with DFO and FeSO_4_. Forward scatter (FSC) data represent the cell size, and side scatter (SSC) data represent the complexity or granularity of the cell. (**f**) The mitochondrial membrane potential by representative frequency histogram ′f 3,3′-dihexyloxacarbocyanine iodide fluorescence (FL1-H). (**g**) Cell cycle analysis by DNA histograms with FL2 dot-plot. M1: sub G1, M2: G0/1, M3: S, M4: G2/M phase. CTL: control. Data represent mean ± SEM. * *p* < 0.05.

**Figure 5 cells-11-02853-f005:**
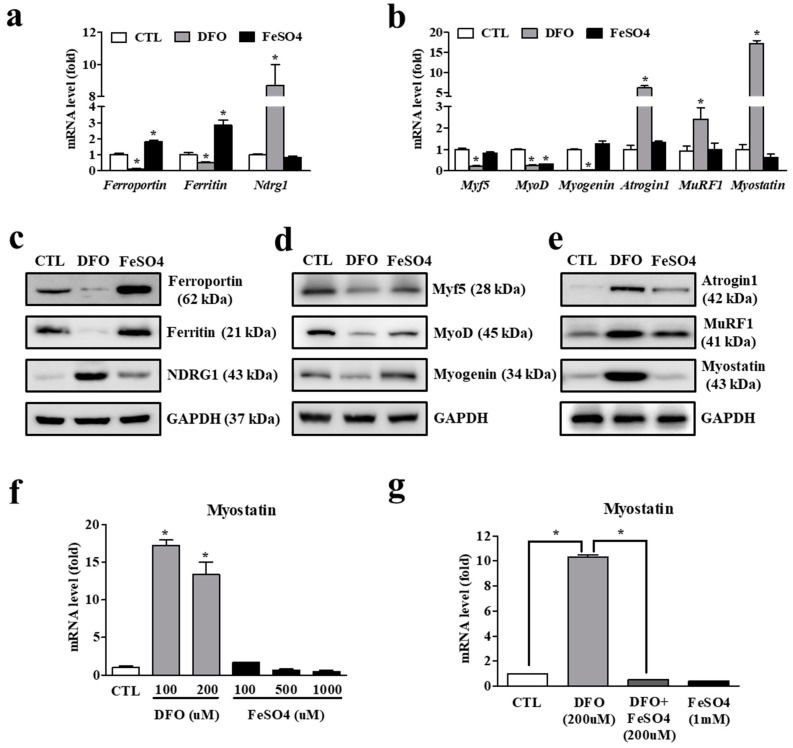
Muscle atrophy-associated gene expression under iron deficiency in myocytes. (**a**) mRNA expression of ferroportin, ferritin and N-Myc downstream regulated 1 (NDGR1) in C2C12 myotubes treated with deferoxamine (DFO) (100 μM/mL) and FeSO_4_ (400 μM /mL) for 24 h. (**b**) mRNA expression of myogenic and muscle atrophy-associated factors in C2C12 myotubes treated with DFO and FeSO_4_ for 48 h. Myf5: myogenic factor 5; MyoD: myoblast determination protein 1; MuRF1: muscle RING-finger protein-1. (**c**) Protein expression of ferroportin, ferritin and NDGR1 in C2C12 myotubes treated with DFO and FeSO_4_. GAPDH: glyceraldehyde 3-phosphate dehydrogenase. (**d**) Protein expression of myogenic factors in C2C12 myotubes treated with DFO and FeSO_4_. (**e**) Protein expression of muscle atrophy-associated factors in C2C12 myotubes treated with DFO and FeSO_4_. (**f**) Expression of myostatin mRNA in C2C12 myotubes treated with different doses of DFO and FeSO_4_. (**g**) Expression of myostatin mRNA in C2C12 myotubes treated with DFO in the presence or absence of FeSO_4_. Data represent mean ± SEM. * *p* < 0.05.

**Figure 6 cells-11-02853-f006:**
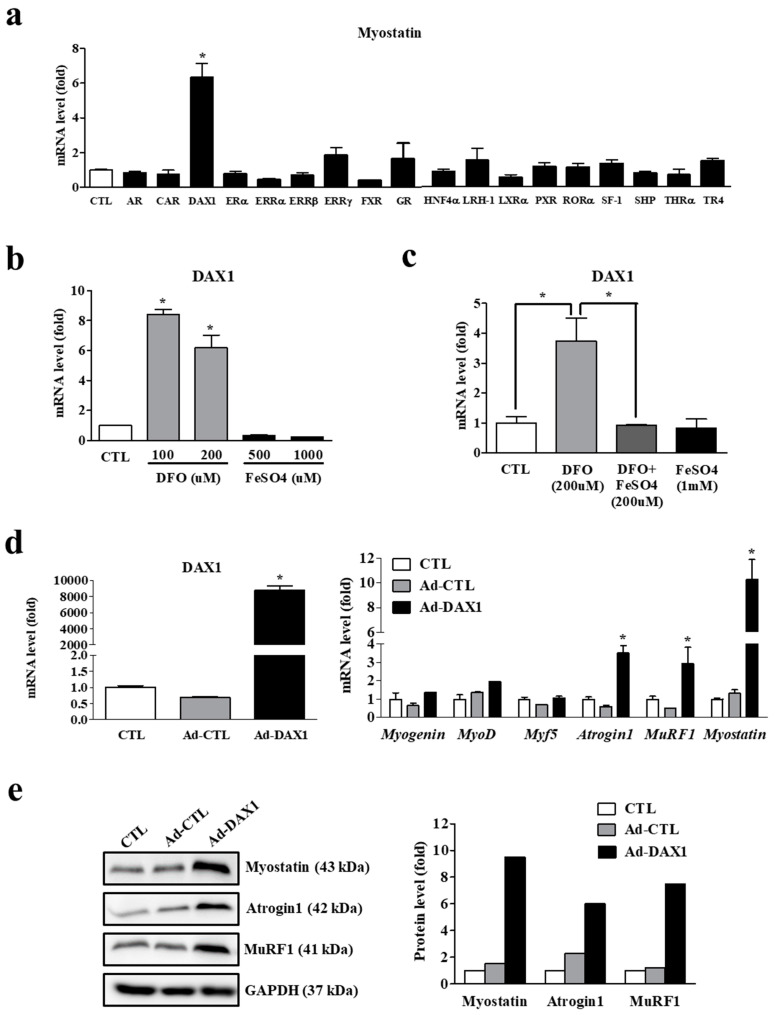
Iron deficiency-induced myostatin regulation by DAX1. (**a**) Expression of myostatin mRNA in C2C12 myotubes transfected with a nuclear receptor for 48 h. AR: androgen receptor; CAR: constitutive androstane receptor; DAX1: dosage-sensitive sex reversal, adrenal hypoplasia congenita critical region, on chromosome X, gene 1; ERα: estrogen receptor α; ERRα/β/γ: estrogen-related receptor α/β/γ; FXR: farnesoid X receptor; GR: glucocorticoid receptor; HNF4α: hepatocyte nuclear factor 4 α; LRH-1: liver receptor homolog-1; LXRα: liver X receptor α; PXR: pregnane X receptor; RORα: RAR-related orphan receptor α; SF-1: steroidogenic factor 1; SHP: small heterodimer partner; THRα: thyroid hormone receptor α; TR4: testicular receptor 4. (**b**) Expression of DAX1 mRNA in C2C12 myotubes treated with different doses of deferoxamine (DFO) and FeSO_4_. (**c**) Expression of DAX1 mRNA in C2C12 myotubes with DFO in the presence or absence of FeSO_4_. (**d**) Adenoviral overexpression of DAX1 and mRNA expression of myogenic and muscle atrophy-associated factors. C2C12 myotubes were infected with an adenovirus expressing DAX1 (Ad-DAX 1) or control vector (Ad-CTL) at 100 multiplicity of infection for 48 h. Myf5: myogenic factor 5; MyoD: myoblast determination protein 1; MuRF1: muscle RING-finger protein-1. (**e**) Protein expression of muscle atrophy-associated factors in C2C12 myotubes infected with Ad-DAX 1. GAPDH: glyceraldehyde 3-phosphate dehydrogenase. A densitometric analysis of the Western blots is shown in the graph on the right. Data represent mean ± SEM. * *p* < 0.05.

**Figure 7 cells-11-02853-f007:**
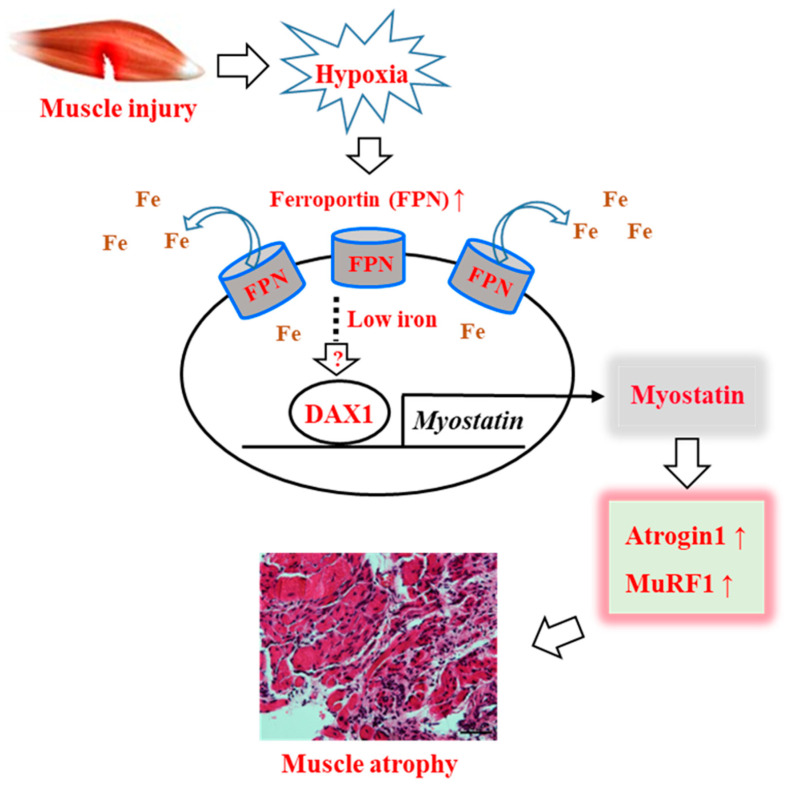
Proposed mechanism for muscle atrophy after muscle injury via the regulation of iron content in myocytes. Muscle injury induces hypoxic conditions, and the activated hypoxia-inducible factor 1α (HIF1α) regulates ferroportin gene transcription, leading to intracellular iron exportation from myocytes. Through an unknown cell-signaling pathway, iron deficiency-induced DAX1 increases myostatin expression, which in turn regulates atrogin1 or MuRF1 expression. CTL: control.

**Table 1 cells-11-02853-t001:** The primer sequences used for qRT-PCR.

Gene Full Name	Gene Symbol	Sequences; Forward (F)/Reverse (R)
(human) Ferritin, heavy polypeptide 1	FTH1	(F) 5′-CCCCCATTTGTGTGACTTCAT-3′ (R) 5′-GCCCGAGGCTTAGCTTTCATT-3′
(mouse) Ferritin, light polypeptide 1	Ftl1	(F) 5′-AATCAGGCCCTCTTGGATCT-3′ (R) 5′-GGCGCTCAAAGAGATACTCG-3′
(human) Hepcidin antimicrobial peptide	HAMP	(F) 5′-ATGGCACTGAGCTCCCAGAT-3′ (R) 5′-CACATCCCACACTTTGATCG-3′
(mouse) Hepcidin antimicrobial peptide	Hamp	(F) 5′-TGCCTGTCTCCTGCTTCTCCT-3′ (R) 5′-GATGGGGAAGTTGGTGTCTC-3′
(human/mouse) Ferroportin	FTN	(F) 5′-CTGCAAGAAAATGTAATTGAATCTG-3′ (R) 5′-ATTGCCACAAAGGAGACTGAAATC-3′
(mouse) Hypoxia inducible factor 1 alpha	Hif1a	(F) 5′-ACCTTCATCGGAAACTCCAAAG-3′ (R) 5′-CTGTTAGGCTGGGAAAAGTTAGG-3′
(mouse) N-myc downstream regulated gene 1	Ndrg1	(F) 5′-ATGTCCCGAGAGCTACATGAC-3′ (R) 5′-CCTGCTCCTGAACATCGAACT-3′
(mouse) Myogenic factor 5	Myf5	(F) 5′-CCTCATGTGGGCCTGCAAAG-3′ (R) 5′-CATTCCTGAGGATCTCCACC-3′
(mouse) Myogenic differentiation 1	Myod1	(F) 5′-CAAGCGCAAGACCACCAACG-3′ (R) 5′-ATATAGCGGATGGCGTTGC-3′
(mouse) Myogenin	Myog	(F) 5′-GAGACATCCCCCTATTTCTACCA-3′ (R) 5′-GCTCAGTCCGCTCATAGCC-3′
(mouse) F-box protein 32 (Atrogin 1)	Fbxo32	(F) 5′-ATGCACACTGGTGCAGAGAG-3′ (R) 5′-TGTAAGCACACAGGCAGGTC-3′
(mouse) Tripartite motif-containing 63 (MuRF1)	Trim63	(F) 5′-ACCTGCTGGTGGAAAACATC-3′ (R) 5′-AGGAGCAAGTAGGCACCTCA-3′
(mouse) Myostatin	Mstn	(F) 5′-AGTGGATCTAAATGAGGGCAGT-3′ (R) 5′-GTTTCCAGGCGCAGCTTAC-3′
(mouse) Nuclear receptor subfamily 0, group B, member 1	Dax1	(F) 5′-AGGGCAGCATCCTCTACAAC-3′ (R) 5′-TGGTCTTCACCACAAAAGCA-3′
(mouse) Glyceraldehyde-3-phosphate dehydrogenase	Gapdh	(F) 5′-ATAATACCGATCCCCGAAGG-3′ (R) 5′-CTGGATGGTGTATGCACAGG-3′
(human/mouse) Actin, beta	Actb	(F) 5′-TCTGGCACCACACCTTCTAC-3′ (R) 5′-TCGTAGATGGGCACAGTGTGG-3′

## Data Availability

Not applicable.

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
