# Peer review of "A Novel Muscle Atrophy Mechanism: Myocyte Degeneration Due to Intracellular Iron Deprivation"

_cells, 2022, doi:10.3390/cells11182853_

Round 1
Reviewer 1 Report
The authors tried to investigate the effects of iron deficiency on skeletal muscle to test their hypothesis that a muscle-injury-associated decrease of intracellular iron in skeletal muscle cells induces muscle atrophy. There are several concerns, however, in their study.
The resolution of several of the images (Figures 2a, 2e and 3e) is fairly low. Finer, higher-resolution images should be added in their place.
Figure 3c:
There are no C2C12 myotubes in the images. The authors only investigated the cell viability of the myoblasts. Myoblasts are undifferentiated precursor muscle cells. Iron deficiency may affect myoblasts differently than it affects the myotubes. The authors should repeat the experiments using myotubes to assess the physiological effects of iron deficiency on skeletal muscle.
Figure 3 shows that not only iron deficiency, but also iron supplementation, degraded the cell viability of C2C12 myoblasts. On the other hand, the experiments demonstrated an upregulation of atrophic genes under an iron-deficient condition, but no such effect with iron supplementation. This discrepancy should be explained.
The authors investigated the iron level in human skeletal muscle. Did the iron level of the muscle tissues reflect the intracellular iron level? If so, the authors should explain their rationale for assessing the tissue iron levels instead of the intracellular iron levels. Did the iron in hemoglobin or interstitial fluid have any effects?
For the experiments demonstrated in Figures 4 and 5, the authors state that they used myotubes. Yet as mentioned above, no myogenic differentiation of the C2C12 cells took place. The authors should repeat these investigations using myotubes.
Dax1 knockdown experiments should be carried out to support the authors’ hypothesis.
Author Response
First of all, we really appreciate for your taking time to review our manuscript with constructive remarks. We definitely agree with your opinion.
The resolution of several of the images (Figures 2a, 2e and 3e) is fairly low. Finer, higher-resolution images should be added in their place.
For a clear distinction in inflammation or muscle fat infiltration due to muscle injury, the picture was taken inevitably in a wide area causing lower resolution. It seems that it is difficult to acquire a higher resolution image with our currently available equipment. We humbly hope you to understand this situation. Instead, we made the figure clearer with some image adjustment because it looked a bit dark as an unmodified image.
Figure 3c: There are no C2C12 myotubes in the images. The authors only investigated the cell viability of the myoblasts. Myoblasts are undifferentiated precursor muscle cells. Iron deficiency may affect myoblasts differently than it affects the myotubes. The authors should repeat the experiments using myotubes to assess the physiological effects of iron deficiency on skeletal muscle.
We definitely agree with your comment.
In fact, in order to compare and to emphasize the shape of the smaller cells in DFO treated C2C12 myotubes (differentiated with DMEM media containing 2% horse serum), we selected the one photo at high magnification. In order to clearly show the differentiation state being myotubes, we re-performed the experiment and changed the photos in Figure 4d. And we confirmed that the differentiated myocytes respond to DFO similarly as shown in previous experiments. (Please see the result in Figures 4b)
Figure 4 shows that not only iron deficiency, but also iron supplementation, degraded the cell viability of C2C12 myoblasts. On the other hand, the experiments demonstrated an upregulation of atrophic genes under an iron-deficient condition, but no such effect with iron supplementation. This discrepancy should be explained.
It has been reported that excess iron accumulation in muscles triggers iron-dependent oxidative stress, producing reactive oxygen species (ROS), indicating another route by which iron imbalance can lead to myocyte damage and skeletal muscle atrophy regardless of any other atrophic gene.
The authors investigated the iron level in human skeletal muscle. Did the iron level of the muscle tissues reflect the intracellular iron level? If so, the authors should explain their rationale for assessing the tissue iron levels instead of the intracellular iron levels. Did the iron in hemoglobin or interstitial fluid have any effects?
It would be desirable to isolate only myocytes from the muscle tissues and measure the iron concentration in the isolated myocytes, but it was difficult to immediately separate myocytes from all muscle specimens collected in the operating room, and there is a possibility that the iron concentration may change during the myocyte separation process. Although we cannot rule out the possibility of iron presence in hemoglobin or interstitial fluid, the measured iron is mostly considered to be iron present in myocytes.
For the experiments demonstrated in Figures 4 and 5, the authors state that they used myotubes. Yet as mentioned above, no myogenic differentiation of the C2C12 cells took place. The authors should repeat these investigations using myotubes.
As mentioned above, C2C12 cells used in the experiment was cultured in DMEM media containing 2% horse serum for 72 hours, and we have confirmed the differentiation into myotubes. (Please see the photos of differentiated cells in Figures 4d)
We definitely agree with the reviewer's opinion. However, I am afraid to say that there is a time constraint to re-perform all the experiments in Figures 4, 5, and 6 again. We are sure that the cell is differentiated obviously.
Dax1 knockdown experiments should be carried out to support the authors’ hypothesis.
We definitely agree with the reviewer's opinion. However, I am afraid to say that there is a time limit for production of siDAX1 or shDAX1 to perform Dax1 knockdown experiments within a given revision period. We would like to perform the experiment as a further study in the future.
Reviewer 2 Report
Suh and colleagues present in their manuscript a novel muscle atrophy mechanism focussing on intracellular iron deprivation. The authors show that damage-induced hypoxic conditions increase the expression of ferroportin, the major iron exporter out of the cell, in myocytes. This causes intracellular iron deficiency which lead to myostatin induction which upregulates atrogin1 en MuRF1 which results in muscle atrophy. Specifically, the authors identified DAX1 as a novel transcriptional regulator of myostatin expression. This is a highly interesting study, very well executed, and which will add important new information to the field. The authors performed extensive, solid experiments and investigated their hypothesis in multiple manners. I just have a couple of comments that I would like to have addressed:
1. In Figure 4A, the authors administer increasing doses of DFO to assess cell vialibility. The cell viability significantly decreases with increases doses of DFO. The authors also showed that increasing doses of FeSO4 (iron overload) influenced less prominently cell viability. However, the authors did not show a model of DFO with concomitantly FeSO4. It would be good to see the effect of DFO + FeSO4 to show that it is really iron deficiency causing less cell viability, and not other metals, e.g., copper or zinc, which could also be affected by DFO and also negatively impact skeletal muscle cells.
2. In Figure 3C, the effect of HIF-1alpha and HIF-1aplha with HIF1blocker is evaluated. Ferroportin is strongly significantly induced. Peculiar enough the increase in HIF-1alpha is rather mild and not significant, do the authors have an explanation for this? And would even more pronounced HIF-1alpha stabilization with HIF-1alpha lead to an even more pronounced spike in ferroportin?
3. Figure 6F is the overall conclusion and proposed pathway by the authors which they nicely investigated. I would suggest to make a separate figure out of this very illustrative figure rather than incorporating it into box F of Figure 6.
Author Response
First of all, we really appreciate for your taking time to review our manuscript with constructive remarks. We definitely agree with the reviewer's opinion.
In Figure 4A, the authors administer increasing doses of DFO to assess cell vialibility. The cell viability significantly decreases with increases doses of DFO. The authors also showed that increasing doses of FeSO4 (iron overload) influenced less prominently cell viability. However, the authors did not show a model of DFO with concomitantly FeSO4. It would be good to see the effect of DFO + FeSO4 to show that it is really iron deficiency causing less cell viability, and not other metals, e.g., copper or zinc, which could also be affected by DFO and also negatively impact skeletal muscle cells.
As recommended by reviewer, we performed the suggested experiment using differentiated C2C12 myotubes, and found out that the decreased cell viability by iron deficiency was significantly recovered by iron supply with concomitant FeSO4 treatment. This result was described in the revised manuscript, and Fig. 4a was modified and new result was added as shown in Fig. 4b.
As for the effects of other metals such as copper or zinc on muscle atrophy, I would gladly accept the reviewer's suggestions, but we couldn’t try to study that because it would be out of scope in light of the topic of iron deficiency.
In Figure 3C, the effect of HIF-1alpha and HIF-1aplha with HIF1blocker is evaluated. Ferroportin is strongly significantly induced. Peculiar enough the increase in HIF-1alpha is rather mild and not significant, do the authors have an explanation for this? And would even more pronounced HIF-1alpha stabilization with HIF-1alpha lead to an even more pronounced spike in ferroportin?
The expression of HIF-1α was induced by transient transfection, and the overexpression level was about 4 times. In general, endogenous HIF-1α in normoxia is present in the cytoplasm, but in case of hypoxic condition, it moves into the nucleus of the cell, binds to HIF-1 β (also called ARNT), and induces target gene expression. In our study, it is speculated that (with HIF-1 β) overexpressed HIF-1α directly regulates the gene expression of ferroportin in the nucleus, indicating a relatively significant ferroportin expression compared to HIF-1α. A significant decrease in ferroportin expression by HIF-1α inhibitor, which suppresses the function of HIF-1α, not HIF-1α expression, suggests that HIF-1α stabilization plays an important role in ferroportin expression.
Figure 6F is the overall conclusion and proposed pathway by the authors which they nicely investigated. I would suggest to make a separate figure out of this very illustrative figure rather than incorporating it into box F of Figure 6.
We appreciate the reviewer's valuable opinion. As recommended by reviewer, we separated Figure 6f as Figure 7 in the revised Figures.
Reviewer 3 Report
In this study the authors analyzed the impact of iron depletion on the induction of muscle atrophy. This relation was studied in cell culture as well as in an animal model and human muscle biopsy samples. The concept described in the present study that iron is decreased after muscle injury due to an increase of ferroportin is very interesting and goes along with iron supplementation studies in heart failure patients. The results of the present study are very interesting and suggest a different model for the induction of muscle atrophy. The manuscript is well written and the results presented are clear and support the presented pathophysiological model. I only have some minor comments.
Comments:
1. A statement is missing that the human and animal study was approved by the local ethnical committee.
2. Method 2.3 iron measurement: was the iron measurement performed in tissue homogenate or plasma. This should be mentioned here
3. Method 2.4: can the author state how the mRNA expression levels were analyzed quantitatively. Please state the method used.
4. In section 2.1 the authors state that all tissue samples were frozen in liquid nitrogen after removal, also the samples for histology. In section 2.6 it is mentioned that for tissue histology paraffin-embedded tissue was used. Please confirm that the tissue was frozen first in N2 before embedding. Otherwise correct.
5. The authors state that for all the analyses a t-test was used. Nevertheless when 3 groups were compared (for example figure 3C, 5A,B) an Anova analysis is required. Please check that always the appropriate statistical test is applied.
6. Labelling the representative western blot with the size of the respective protein would help.
Author Response
First of all, we really appreciate for your taking time to review our manuscript with constructive remarks. We definitely agree with the reviewer's opinion.
A statement is missing that the human and animal study was approved by the local ethnical committee.
It was stated separately in the ‘Ethical Guidelines Statement’ after ‘Discussion’ section.
Method 2.3 iron measurement: was the iron measurement performed in tissue homogenate or plasma. This should be mentioned here
Muscle tissue or intracellular iron concentration was measured from tissue homogenate or harvested cells, respectively. As recommended by reviewer, we described this issue in ‘Method 2.3’.
Method 2.4: can the author state how the mRNA expression levels were analyzed quantitatively. Please state the method used.
We are sorry for the confusing description in the ‘method section’. Reverse transcription quantitative real-time PCR (RT-qPCR) is a sensitive technique for the quantification of steady-state mRNA levels. The process is performed by reverse transcription of total RNA or mRNA to complementary DNA (cDNA) by the enzyme reverse transcriptase, followed by amplification and detection of specific targets of this cDNA using a technique called quantitative PCR (qPCR) or real-time PCR. At each cycle during this PCR, the quantity of DNA is measured in real-time by using a fluorescent chemistry. Because the fluorescence signal increases proportionally to the amount of replicated DNA, the DNA is quantified in “real time”, and the resultant PCR product concentration is directly proportional to the initial starting quantity of mRNA.
We corrected the subtitle of method as a standardized acronym in the revised manuscript.
In section 2.1 the authors state that all tissue samples were frozen in liquid nitrogen after removal, also the samples for histology. In section 2.6 it is mentioned that for tissue histology paraffin-embedded tissue was used. Please confirm that the tissue was frozen first in N2 before embedding. Otherwise correct.
We are sorry for the confusing description in the ‘method section’.
The collected muscle samples were immediately frozen at -80 °C for RT-qPCR and western blot analyses or were fixed in fresh 4% buffered formalin for 24 hours at 4°C for tissue histology such as hematoxylin and eosin (H&E) staining.
We corrected this issue in ‘Method 2.3’ of the revised manuscript.
The authors state that for all the analyses a t-test was used. Nevertheless when 3 groups were compared (for example figure 3C, 5A, B) an Anova analysis is required. Please check that always the appropriate statistical test is applied.
Because we compared the differences between control and experimental groups, individually like DFO per control (DFO/CTL) and FeSO4 per control (FeSO4/CTL), t-test was used as a statistical analysis. Control is 1 as a baseline.
Labelling the representative western blot with the size of the respective protein would help.
As recommended by reviewer, we denoted the protein size in the revised figures.
Round 2
Reviewer 1 Report
The authors made few revisions to their manuscript since the last submission. As such, the most serious concerns pointed out earlier remain. The authors should replace the histological images and carry out most of the analyses again using C2C12 myotubes.
As I pointed out earlier, the resolution of several of the images (Figures 2a, 2e and 3e) remains fairly low. The authors should present high-resolution images in a simple, straightforward manner.
Figure 4d:
The original manuscript clearly presented the C2C12 myoblasts as C2C12 myotubes. I took this as a strong indication that all of the data were derived from C2C12 myoblasts. The authors should clearly explain why they presented such images in the original manuscript.
The revised manuscript presents images of C2C12 myotubes instead of C2C12 myoblasts, in response to my previous comments. On the other hand, the other data from the C2C12 cells match the data shown in the original Figures 4, 5, and 6. As I pointed out in my previous review, iron deficiency may affect myoblasts differently than it affects the myotubes. The authors should repeat all of the evaluations and analyses using myotubes to assess the physiological effects of iron deficiency on skeletal muscle. The population of undifferentiated myoblasts in the sample from differentiated C2C12 myotubes, for example, is small. Is the number of cells in Figure 4 reasonable? The cells composing the myotubes are too big. The authors state that they analyzed the cell cycle of the myotubes using flow cytometry. Is this true? Can flow cytometry be used to characterize myotubes in this way? Myotubes have multinucleated cells. How did the authors determine the cell cycle of the multinucleated myotubes? Myoblasts, but not myotubes, can be characterized by flow cytometry.
Further, the physiological characteristics of myotubes depend on the differentiation level. At the very least, the authors should analyze and show the fusion and/or differentiation index to ensure the reproducibility of their experiments.
Most myotubes at 100 uM DFO were detached from the culture plate. This myotube detachment may be a sign of cell death. If cell death occurred, could it have been caused by the cytotoxicity of the DFO?
I believe that Dax1 knockdown experiments would provide essential evidence and support for this study, as I pointed out in my previous review.
Author Response
As I pointed out earlier, the resolution of several of the images (Figures 2a, 2e and 3e) remains fairly low. The authors should present high-resolution images in a simple, straightforward manner.
In the case of 'Fig. 2a and 3e', the pictures were taken to compare the occurrence of fat deposition and inflammation by muscle damage in a wide view. At present, due to the limitations of our currently available equipment, it is difficult to take more high-resolution of photos. In particular, in the case of the mouse tissue in 'Fig. 3e', it is difficult to obtain the same tissue and slide, because the experiments were performed at another institution few years ago. Therefore, it takes more time to reproduce the same experiment and to take same photos again.
We humbly ask for your generous understanding as long as there is no significant impediment to understanding the overall result.
Figure 4d:
The original manuscript clearly presented the C2C12 myoblasts as C2C12 myotubes. I took this as a strong indication that all of the data were derived from C2C12 myoblasts. The authors should clearly explain why they presented such images in the original manuscript.
The revised manuscript presents images of C2C12 myotubes instead of C2C12 myoblasts, in response to my previous comments. On the other hand, the other data from the C2C12 cells match the data shown in the original Figures 4, 5, and 6. As I pointed out in my previous review, iron deficiency may affect myoblasts differently than it affects the myotubes. The authors should repeat all of the evaluations and analyses using myotubes to assess the physiological effects of iron deficiency on skeletal muscle. The population of undifferentiated myoblasts in the sample from differentiated C2C12 myotubes, for example, is small. Is the number of cells in Figure 4 reasonable? The cells composing the myotubes are too big. The authors state that they analyzed the cell cycle of the myotubes using flow cytometry. Is this true? Can flow cytometry be used to characterize myotubes in this way? Myotubes have multinucleated cells. How did the authors determine the cell cycle of the multinucleated myotubes? Myoblasts, but not myotubes, can be characterized by flow cytometry.
Further, the physiological characteristics of myotubes depend on the differentiation level. At the very least, the authors should analyze and show the fusion and/or differentiation index to ensure the reproducibility of their experiments.
We apologize for any misunderstanding. As explained in the 1st-revision process, we clearly had been used the differentiated C2C12 in the corresponding all experiments. However, with respect to the figures in original manuscript, we intentionally selected the photos to show the shape of cells revealing clear cell shrinkage and reduced cell population caused by DFO. Although we couldn't rule out the existence of some residual myoblasts in culture dish, we confirmed obviously the induction of C2C12 differentiation into myotubes in the currently revised experiments, and also confirmed the cell responded properly by DFO again. Please understand the situation that it is rather difficult to reproduce all the corresponding experiments again for a given revision period.
Most myotubes at 100 uM DFO were detached from the culture plate. This myotube detachment may be a sign of cell death. If cell death occurred, could it have been caused by the cytotoxicity of the DFO?
By flow cytometry analysis, a decrease in the FSC and an increase in SSC under iron deprivation by DFO treatment indicate a substantial cell shrink and apoptosis. Moreover, DFO treatment also led to significant depleted mitochondrial membrane potential as well as cell cycle arrest. Taken together, it is speculated that cell death is induced by these various causes.
I believe that Dax1 knockdown experiments would provide essential evidence and support for this study, as I pointed out in my previous review.
We definitely agree with your comment. However, it seems rather difficult to prepare siDAX1 or shDAX1 and perform Dax1-knockdown experiments within a given revision period.
Reviewer 2 Report
My previous comments have been well addressed, no further comments.
Author Response
We really appreciate for your taking time to review our manuscript with constructive remarks.